# De novo modeling of the $F_{420}$-reducing [NiFe]-hydrogenase from a methanogenic archaeon by cryo-electron microscopy

**Deryck J Mills[1†], Stella Vitt[2†], Mike Strauss[1‡], Seigo Shima[2,3], Janet Vonck[1*]**

[1]Department of Structural Biology, Max Planck Institute of Biophysics, Frankfurt, Germany; [2]Max Planck Institute for Terrestrial Microbiology, Marburg, Germany; [3]PRESTO, Japan Science and Technology Agency, Kawaguchi, Japan

**Abstract** Methanogenic archaea use a [NiFe]-hydrogenase, Frh, for oxidation/reduction of $F_{420}$, an important hydride carrier in the methanogenesis pathway from $H_2$ and $CO_2$. Frh accounts for about 1% of the cytoplasmic protein and forms a huge complex consisting of FrhABG heterotrimers with each a [NiFe] center, four Fe-S clusters and an FAD. Here, we report the structure determined by near-atomic resolution cryo-EM of Frh with and without bound substrate $F_{420}$. The polypeptide chains of FrhB, for which there was no homolog, was traced de novo from the EM map. The 1.2-MDa complex contains 12 copies of the heterotrimer, which unexpectedly form a spherical protein shell with a hollow core. The cryo-EM map reveals strong electron density of the chains of metal clusters running parallel to the protein shell, and the $F_{420}$-binding site is located at the end of the chain near the outside of the spherical structure.

**\*For correspondence:**
janet.vonck@biophys.mpg.de

[†]These authors contributed equally to this work

[‡]**Present address:** Department of Biological Chemistry and Molecular Pharmacology, Harvard Medical School, Boston, MA, United States

**Competing interests:** The authors declare that no competing interests exist

**Reviewing editor**: Wes Sundquist, University of Utah, United States

## Introduction

[NiFe]-hydrogenases are microbial enzymes that heterolytically cleave $H_2$, after which the electrons from a hydride ion are reversibly transferred to electron carriers. These enzymes are involved in many metabolic pathways in microbial ecosystems, notably in methanogenesis. The members of the [NiFe]-hydrogenase family are composed of at least two subunits, large (~60 kDa) and small (~30 kDa), which contain a [NiFe] dinuclear metal center and three iron–sulfur (Fe-S) clusters, respectively. The family is classified into five distantly related groups based on a phylogenetic analysis of their primary structures (*Vignais and Billoud, 2007*; *Constant et al., 2011*). The cytoplasmic $F_{420}$-reducing [NiFe]-hydrogenase (Frh) belongs to the group 3 [NiFe]-hydrogenases, which catalyze the reversible reduction of the soluble hydride carrier, NAD(P) or $F_{420}$. $F_{420}$ is a deazaflavin derivative that acts as an important hydride acceptor/donor in the central methanogenic pathway. The group 3 hydrogenases contain an additional subunit that interacts with the soluble coenzymes, for example, the iron–sulfur flavoprotein FrhB. To date, the crystal structures of [NiFe]-hydrogenases only from group 1 have been determined: six derived from sulfate-reducing bacteria, including two orthologs harboring a [NiFeSe]-center as active site (*Volbeda et al., 1995*; *Higuchi et al., 1997*; *Montet et al., 1997*; *Garcin et al., 1999*; *Matias et al., 2001*; *Marques et al., 2010*), one from a photosynthetic bacterium (*Ogata et al., 2010*), and three from hydrogen-oxidizing bacteria and *Escherichia coli*, which in contrast to the other species are oxygen-tolerant and have an usual [4Fe-3S] proximal cluster (*Fritsch et al., 2011*; *Shomura et al., 2011*; *Volbeda et al., 2012*).

Frh is a key enzyme in the metabolism of methanogenic archaea (*Thauer et al., 2010*). The reduction of carbon dioxide to methane is an overall eight-electron reduction process involving the oxidation of four molecules of $H_2$ by several hydrogenases. Four electrons are provided through the reduced form of the coenzyme $F_{420}$, which is regenerated by Frh. Frh is a heterotrimeric enzyme composed of the

**eLife digest** Many microbes grow by producing methane gas from carbon dioxide and hydrogen gas, and enzymes known as hydrogenases play important roles in this metabolic process. The production of methane in these microbes depends on a nickel–iron hydrogenase called Frh adding electrons to a coenzyme called $F_{420}$. This hydrogenase cleaves a hydrogen molecule into two electrons, which are transferred to the $F_{420}$ coenzyme, and two protons. The reduced form of $F_{420}$ is then used for several reactions in the methane production process. This process, which is known as methanogenesis, provides the microbes with energy.

Nickel–iron hydrogenases can be divided into five different groups, but researchers have been able to determine the detailed structures of the enzymes in just one of these groups. All nickel–iron hydrogenases contain at least two subunits: a large subunit with a catalytic center composed of both nickel and iron ions and a small subunit that contains three iron–sulfur clusters. Frh—which is short for $F_{420}$-reducing nickel–iron hydrogenase—is known to have a third subunit comprising an extra iron–sulfur cluster and a coenzyme called FAD that allows it to interact with the $F_{420}$ coenzyme. However, until now, little was known about the detailed structure of the Frh enzyme.

Mills et al. have used electron cryo-microscopy (cryo-EM) to determine the structure of Frh when it is on its own, and also when it is bound to $F_{420}$. This technique involves freezing a solution of the enzyme in a thin layer of ice and recording an image of this layer in an electron microscope. By combining a large number of images, each of which contains many identical enzymes in different orientations, it is possible to determine the 3-dimensional structure of the enzyme.

Mills et al. found that Frh forms a very large tetrahedral complex that contains six Frh dimers. And by comparing the structure with and without $F_{420}$, they identify a pocket near the FAD coenzyme that the $F_{420}$ coenzyme binds to. They also identify a fold in the third subunit that allows proteins to bind both FAD and $F_{420}$. The work demonstrates the potential of cryo-EM to elucidate structures that cannot be determined by other approaches.

large subunit FrhA (45 kDa) with a binuclear [NiFe]-center, the small subunit FrhG (26 kDa) with three [4Fe4S] clusters and the iron–sulfur flavoprotein FrhB (31 kDa) with a [4Fe4S] cluster and FAD, and the $F_{420}$-binding site. Frh accounts for about 1% of the cytoplasmic protein and has been shown by electron microscopy to form a huge complex of similar appearance in all species investigated: *Methanococcus voltae* (*Muth et al., 1987*), *Methanospirillum hungatei* (*Sprott et al., 1987*), *Methanobacterium thermoautotrophicum* ΔH (*Wackett et al., 1987*), and *Methanobacterium thermoautotrophicum* Marburg (*Braks et al., 1994*). From negative stain images, the complex was interpreted as a flat cylinder consisting of an octamer (*Wackett et al., 1987*) or hexamer (*Braks et al., 1994*) of the heterotrimer.

Recently, developments in instrumentation and image processing software have made it possible to determine structures of large macromolecular complexes to near-atomic resolution by cryo-electron microscopy. Most successful have been studies of large icosahedral viruses (*Yu et al., 2008*, *2011*; *Liu et al., 2010*; *Wolf et al., 2010*; *Zhang et al., 2010a*; *Chen et al., 2011*; *Settembre et al., 2011*), but also a few smaller complexes in the 1 MDa size range of lower symmetry have yielded high-resolution structures, notably GroEL (*Ludtke et al., 2008*) and archaeal chaperonins (*Cong et al., 2010*, *2012*; *Zhang et al., 2010b*). We present here the structure of the Frh complex from the hydrogenotrophic methanogenic archaeon, *Methanothermobacter marburgensis,* by cryo-electron microscopy. The Frh complex is a 1.2-MDa dodecamer with tetrahedral symmetry. The location of all cofactors were identified, and the backbone of the three proteins was traced, including FrhB that has a novel fold.

## Results

The Frh complex was highly purified from *M. marburgensis* under strict anaerobic conditions in the presence of FAD. Its homogeneity was confirmed by electron microscopy with negative staining. We found that all projections of the Frh complex are ring-shaped with a diameter of ~16 nm (*Figure 1A*). Class averages showed twofold and threefold symmetry (*Figure 1B*), which is consistent with tetrahedral symmetry. We conclude that the complexes consist of six dimers of FrhABG arranged into a shell around a hollow core, not in a cylindrical arrangement as in previous EM models (*Wackett et al., 1987*; *Braks et al., 1994*). The dodecamer has a total MW of 1.2 MDa. Iterative structural refinement of a

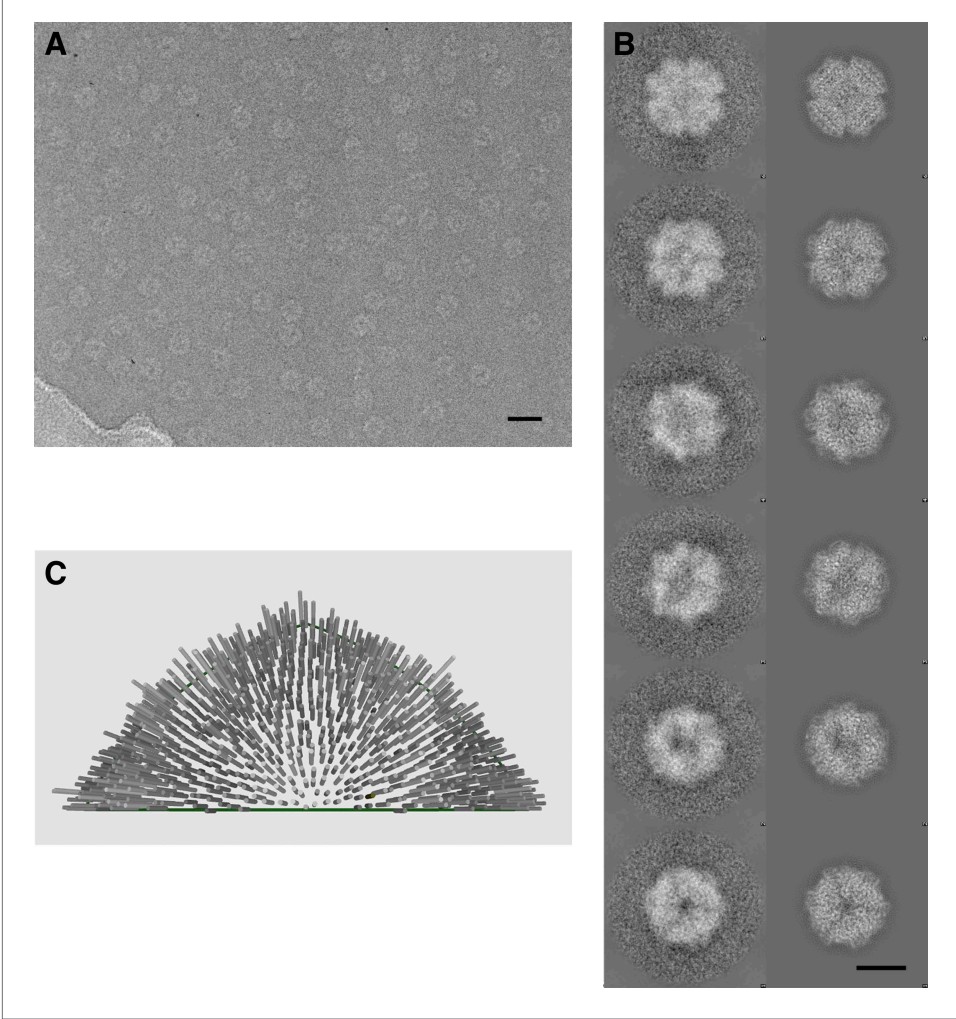

**Figure 1**. Cryo-electron microscopy and image processing. (**A**) A representative area of an electron micrograph taken at 200 kV on an FEI Polara. The defocus was determined as 2.05 μm. The scale bar represents 25 nm. (**B**) Representative class averages and corresponding reprojections of the model for the final refinement iteration. The first and last images represent a view down the twofold and threefold axis of the tetrahedron, respectively. The scale bar represents 10 nm. (**C**) Euler angle distribution for the final reconstruction. Each cylinder represents one class average in the asymmetric triangle (1/12 of the tetrahedron); the height of the cylinder is proportional to the number of particles in the class. The equal distribution shows that there are no preferred orientations for the Frh complex.

84,000-particle cryo-EM dataset, applying tetrahedral symmetry, resulted in a 3D reconstruction (*Figure 2*) at a resolution where the pitch of α-helices and many side chains are recognizable in the map, and β-strands are clearly separated (*Figure 2E–G, 5B*). The globular protein complex showed no preferred orientations in the ice, resulting in a homogeneous coverage of angles (*Figure 1C*). The map shows six distinct dimers, all clearly separated from one another (*Figure 2C*), sitting on the faces of a cube. Each monomer contains a series of five high-density features separated by ~10 Å, forming an arc in the interior of the protein (*Figure 2D* and *Video 1*). These features are easily assigned to the four Fe-S clusters and the [NiFe] center; the latter is recognized by its smaller size. Although at first glance the position of the FrhABG heterotrimers in the dimer is not obvious, the path of the electron transfer chains provides definitive clues to the location of an FrhABG heterotrimer. FrhABG contains one approximately 35-Å-long helix on the outer surface and a bundle of four up to 40-Å-long helices at the interface of different dimers, running from the outside to the interior of the complex. Other helices are considerably shorter.

The large subunit FrhA has considerable homology to the known large subunit structures (*Figure 3*), and is characterized by four long α-helices. Docking the large subunit of the group 1 enzyme into the

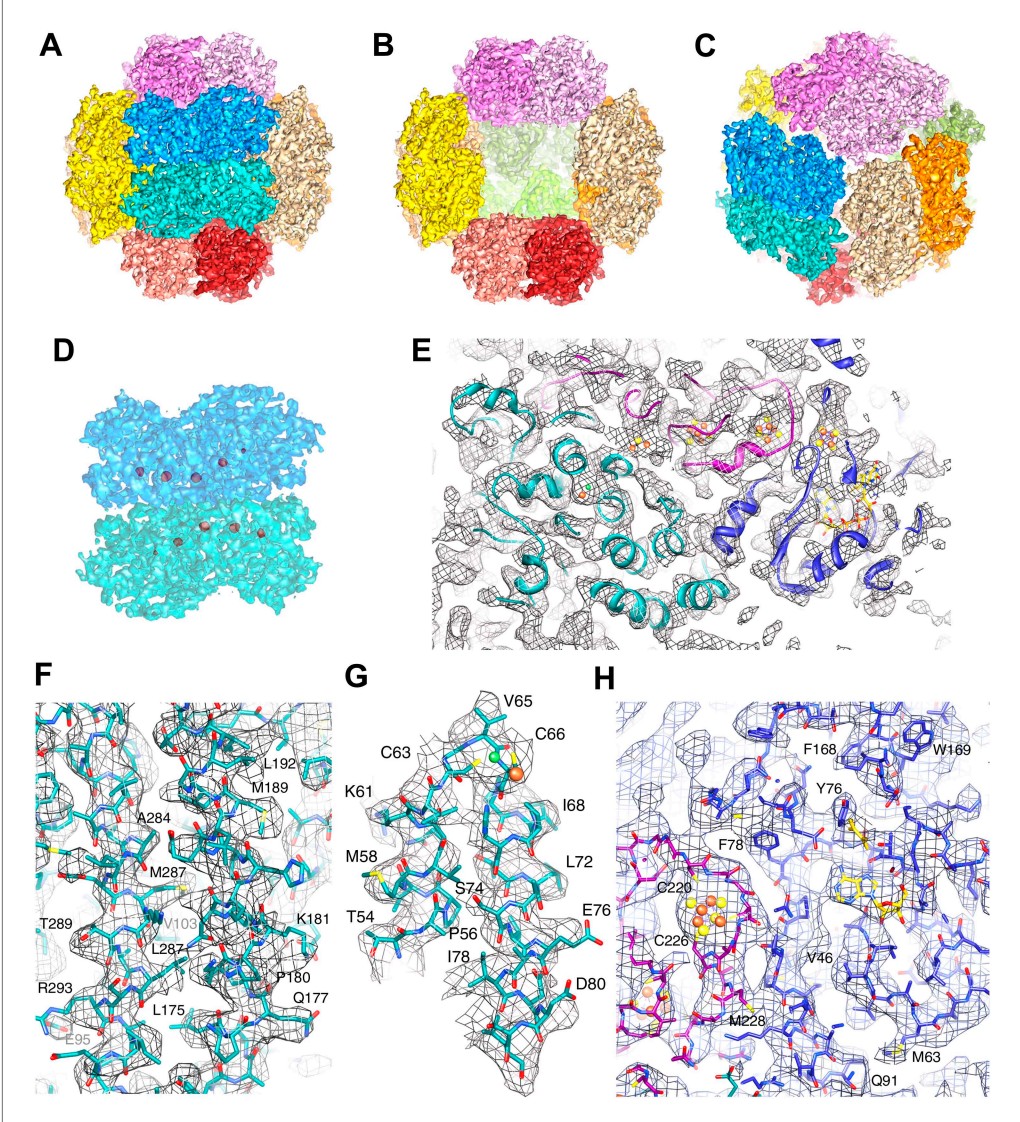

**Figure 2**. High-resolution cryo-EM map of Frh. (**A**) view down the twofold axis. Each of the 12 FrhABG heterotrimers is shown in a different color. (**B**) The same view as (**A**) with the two trimers at the front removed, (**C**) view down the threefold axis. (**D**) Close-up of the front dimer in (**A**) shown as a transparent surface with the high densities of the metal centers in red. (**E**) A 10-Å thick slice of the map with a ribbon model for one FrhABG subunit (the green one in **A** and **D**) superimposed. In this and other figures, FrhA is green, FrhG purple, and FrhB slate-blue. At this level, a complete chain of cofactors can be seen: the [NiFe] cluster in FrhA (green and brown spheres), three FeS clusters in FrhG and one in FrhB (brown and yellow spheres), and the FAD in FrhB (yellow sticks). Details of the map with a full-atom model superimposed: (**F**) Part of the α-helix bundle in FrhA, (**G**) Helix hairpin in FrhA coordinating the [NiFe] center (green and brown balls), (**H**) Part of FrhG and FrhB showing an FeS cluster and FAD. In (**E–H**), the map was filtered at 4.5-Å resolution and sharpened using a B-factor of -54 Å².

map provides a good fit for the α-helix bundle, with the [NiFe] center correctly situated in its identified location and many other structural elements of FrhA matching closely, including two three-stranded antiparallel β-sheets. FrhA is about 15 kDa smaller than the large subunits of known structure; all missing elements are located on the periphery. A structural model for FrhA (***Figure 4A***) was created based on this fit and a sequence alignment with the protein from *Desulfovibrio vulgaris* Hildenborough (***Figure 3***). All but the first three amino acids in the sequence could be traced in our maps. Densities for large side chains confirm the structural assignment (***Figures 2, 5*** and ***Video 2***). The [NiFe] center is located near four cysteine residues, Cys 63, 66, 380, and 383, which are fully conserved in

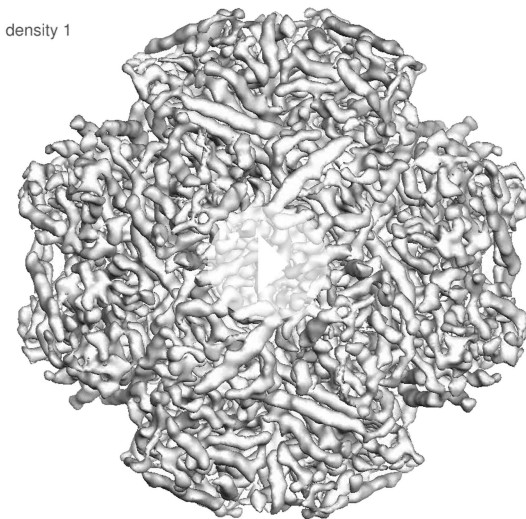

density 1

**Video 1**. The Frh map at different density levels. The Frh map contains high densities corresponding to the metal clusters. The map, filtered to 5.5 Å for clarity, is shown at increasing isosurface levels. At density level 1 (gray), the whole complex is seen; levels 2 (yellow) and 3 (gold) show decreasing protein density. At level 4 (orange), only a string of five densities remains, corresponding to the four [4Fe4S] clusters and the [NiFe] cluster. The FeS clusters, but not the [NiFe], are still visible at level 5 and 6 (red).

all [NiFe]-hydrogenases. The group 1 [NiFe]-hydrogenases contain another ion, typically either $Mg^{2+}$ or $Fe^{2+/3+}$ (*Higuchi et al., 1997*; *Garcin et al., 1999*). In our map, there is strong density at this position (*Figure 5D*), which is part of the structurally conserved region of FrhA, and its surrounding residues, Glu44, Glu229, and His386, are fully conserved in FrhA (*Figure 3A*), so we interpret this density as an ion. One of the helices of the four-helix bundle has a pronounced kink. This helix was interpreted as residues 166–193, and a proline residue (Pro180) not present in the group I proteins was found exactly at the position of the kink, supporting the assignment. No density is present beyond His386, and there is no room in the structure for the final 19 amino acids of the protein. The C-terminal residues after the equivalent histidine in [NiFe]-hydrogenases are enzymatically removed during maturation of the enzymes (*Menon et al., 1993*; *Theodoratou et al., 2005*). We can conclude that FrhA is posttranslationally modified in the same way, likely by FrhD, a homologous endopeptidase contained in the Frh operon FrhADGB (*Liesegang et al., 2010*; *Thauer et al., 2010*).

The fit of the small subunit of the group 1 enzyme reveals structural homology to the N-terminal domain of FrhG including the proximal Fe-S cluster. Some peripheral helices present in the group 1 protein are missing in the FrhG structure, and the similarity is generally lower than for the large subunit (*Figure 6C*). The N-terminal domain is characterized by a conserved parallel β-sheet flanked by short α-helices (*Figure 4B*). In the group 1 [NiFe]-hydrogenases, the proximal Fe-S cluster is located at the C-terminal end of the β-sheet and is coordinated by four cysteines. In the primary structure of FrhG, one of the conserved cysteine residues for the coordination of this cluster is replaced by Asp60 (*Figure 6A*). The bacterial hydrogenases as well as most of the archaeal species have a cysteine at this position, but aspartates have been identified before as ligands of [4Fe4S] clusters (*Calzolai et al., 1995*; *Muraki et al., 2010*; *Gruner et al., 2011*). Accordingly, we positioned this aspartate near the proximal Fe-S cluster.

The C-terminal domain of FrhG has no homology to the group 1 [NiFe]-hydrogenases. Instead, the sequence indicates a ferredoxin domain, with two CxxCxxCxxxC sequences coordinating two [4Fe4S] clusters (*Figure 6A*) (*Alex et al., 1990*). Accordingly, we modeled the C-terminal domain of FrhG on ferredoxin (*Figures 4B, 5A* and *Video 3*). In our model, the residues 235–244 form a β-hairpin with Gly240 at the turn. The two β-strands were predicted and the hairpin is clearly visible in the density, confirming the assignment. Unlike FrhA, which could be traced over its full length, density of FrhG for a 12-residue stretch (188–199) is missing (*Figure 6A*). This region is located at the surface and may be disordered. FrhG as deduced from the *M. marburgensis* genome sequence is 275 residues long (*Liesegang et al., 2010*), but no density is present for the 45 N-terminal amino acids. The *frhG* gene most probably starts with the initiation codon GTG annotated as Val39 in the genome sequence (*Fox et al., 1987*; *Alex et al., 1990*), which means that density is missing in the map for only the first seven amino acids of FrhG.

FrhB contains one Fe-S cluster, an FAD, and the $F_{420}$-binding site. There is no homolog of known structure. FrhB was therefore traced ab initio, aided by secondary structure predictions (*Figure 7*) and recognizable features in the map that included 10 α-helices, 11 β-strands, and the Fe-S cluster. The most prominent feature of FrhB is a ~35 Å surface helix that corresponds to the predicted 24-residue C-terminal helix.

We examined the $F_{420}$-binding proteins of known structure in the protein databank, and found no structural homology with FrhB. All these proteins have cofactors different from FAD, for example,

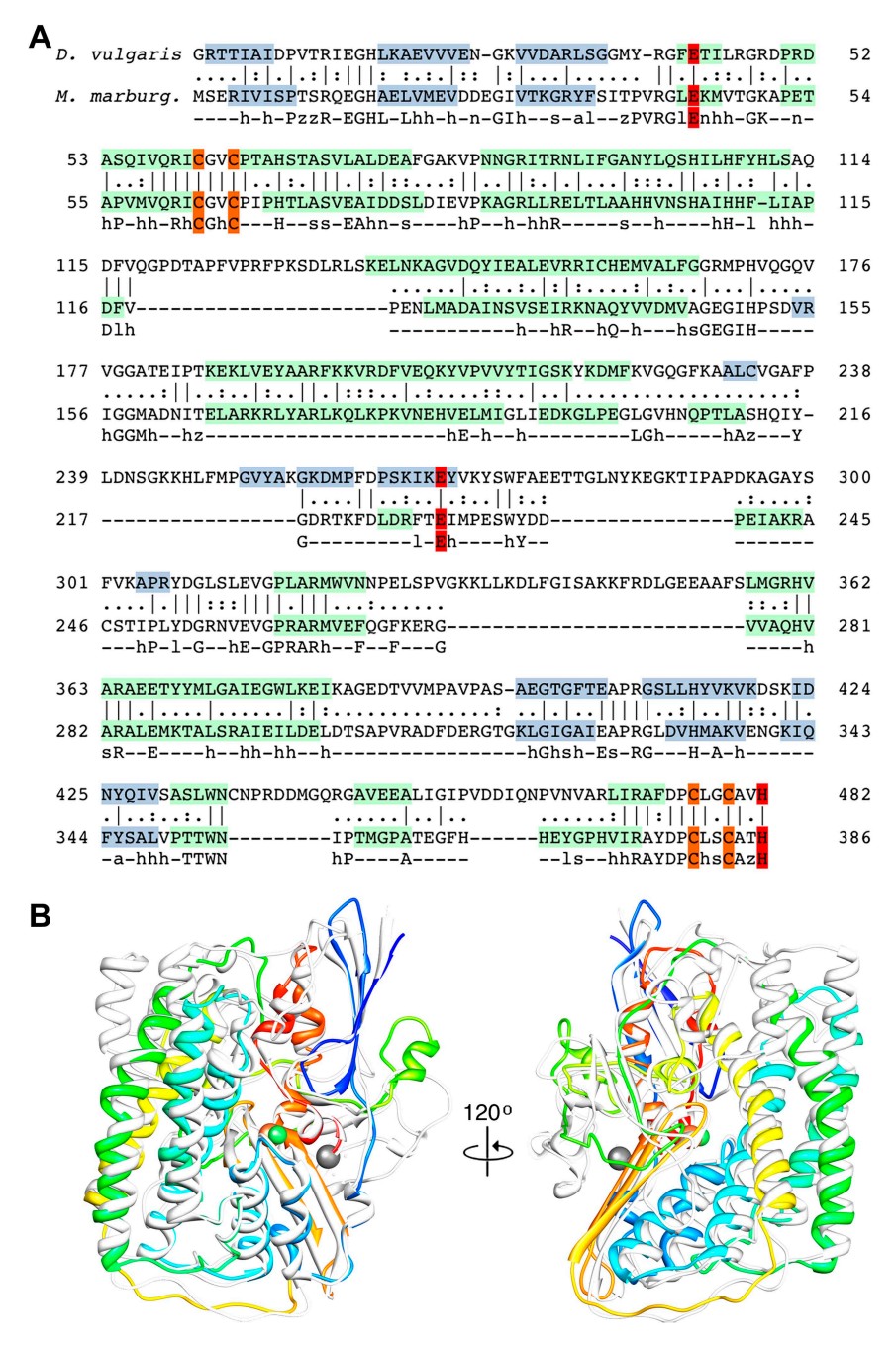

**Figure 3**. Sequence and secondary structure of FrhA and a comparison with homologous proteins. (**A**) Alignment of the [NiFe]-hydrogenase large subunit from *Desulfovibrio vulgaris* Hildenborough (first line) with *Methanothermobacter marburgensis* FrhA (third line). The second line shows the identical (|) and similar (:) amino acids. The alignment was done with EMBOSS needle (http://www.ebi.ac.uk/Tools/psa/emboss_needle/) and manually adjusted after fitting the FrhA structure. The fourth line shows the consensus sequence of 19 archaeal FrhA species. Identical amino acids in capitals, similar ones in lower case (h: hydrophobic; s: small (GAS); l: large (LIFYHW); a: aromatic (FYWH); z: T or S; n: negative, D or E; p: positive, R or K). The α-helices are highlighted in green, β-strands in blue. The [NiFe] ligands are in orange in the ligands of the third ion in red. (**B**) Comparison of the FrhA model (rainbow coloring from blue to red) and the group 1 [NiFe]-hydrogenase large subunit from *Desulfovibrio vulgaris* Hildenborough (gray) (pdb 2wpn) (***Marques et al., 2010***) in two different orientations (left and right). The [NiFe] center and another ion (spheres) overlap as do the structural elements around them, as well as two 3-stranded β-sheets and the lower part of the four-helix bundle. Differences are confined to the periphery.

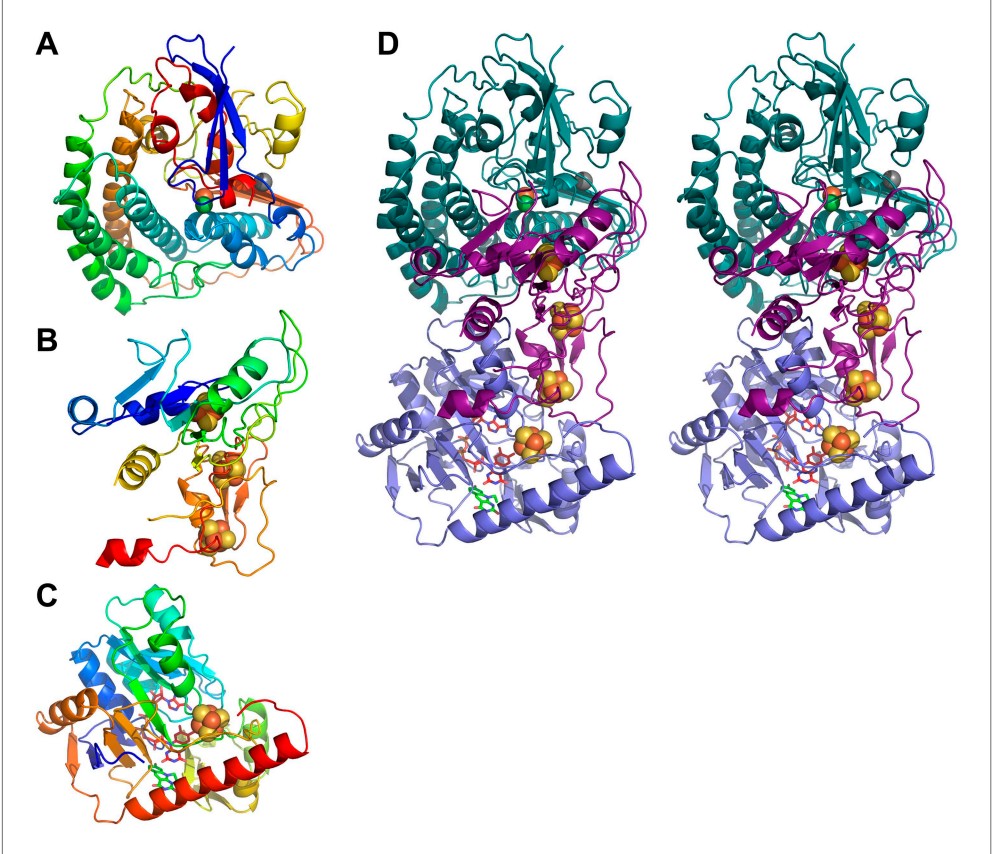

**Figure 4**. Models of Frh subunits. (**A**) FrhA, (**B**) FrhG, and (**C**) FrhB in rainbow colors from blue (N-terminus) to red (C-terminus). (**D**) Stereo pair of the FrhABG heterotrimer with the string of four bound Fe-S clusters (orange/yellow), the [NiFe] binuclear center (green/orange) at the top, and the FAD (red) and $F_{420}$ cofactors (green) below. FrhA is green, FrhG is purple, and FrhB is slate-blue.

methylene-$H_4$MPT or NADP (***Ceh et al., 2009***). Also no homologous FAD-binding proteins were found. We conclude that FrhB has a novel fold (***Figure 4C*** and ***Video 4***). The Fe-S cluster is most probably a 4Fe-4S cluster because it is surrounded by four cysteines (Cys 104, 134, 192, and 195), which are all fully conserved (***Figure 7***). The core of FrhB is a mixed six-stranded β-sheet. A density located ~8 Å from the Fe-S cluster could be interpreted as the isoalloxazine ring of the FAD cofactor. The pyrophosphate moiety is located at the N-terminal end of an α-helix, which is a common FAD-binding motif where the dipole of the helix compensates the negative charge (***Dym and Eisenberg, 2001***) (***Figure 8***), and nearby density for the adenine moiety is visible. The FAD has a bent conformation, with the adenine and isoalloxazine ring only 4–5 Å apart (***Figure 8***) near the highly conserved loop Lys75-Tyr76. The FAD is completely surrounded by highly conserved protein regions (***Figure 7*** and ***Video 5***), further verifying the backbone trace.

To identify the substrate-binding site, we imaged the FrhABG complex in the presence of a large excess of $F_{420}$, and calculated maps, which were refined to a similar resolution. At all refinement stages, the map showed an additional density in a region that was empty in maps of Frh without substrate (***Figure 8A,B***). This density runs approximately parallel to the FAD isoalloxazine ring, at a distance of ~4 Å. We conclude that it represents the isoalloxazine ring of the $F_{420}$ substrate. It is seated in a large pocket surrounded by highly conserved residues, with easy access to the surface of the complex. The $F_{420}$ density is close to the small residues S209 and V210 in a loop between two β-strands, which are part of the longest conserved region of FrhB (***Figure 7***). A predicted three-stranded β-sheet (148–172) flanked by two short helices (***Figure 7***) fits nicely in density. The only conserved region of this sheet, 163IGKGK, forms a turn close to the position of the isoalloxazine ring of $F_{420}$. The location between this ring and the surface of the complex suggests that one or

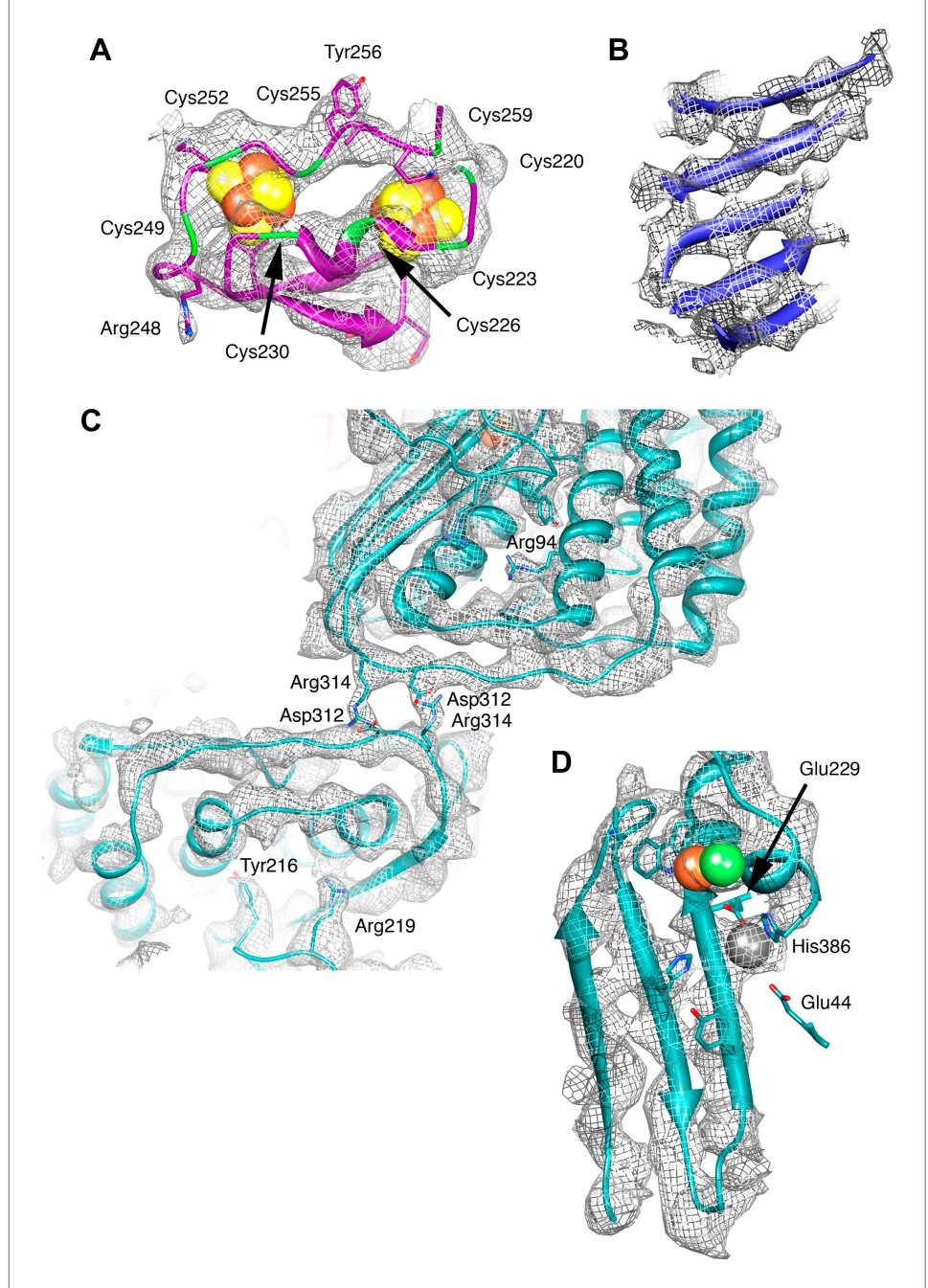

**Figure 5**. Details of the model and map. (**A**) The ferredoxin domain of FrhG. The eight cysteine residues surrounding the two [Fe4S4] clusters are shown in green. (**B**) β-Sheet in FrhB. The strands are clearly separated. (**C**) The dimer interface of FrhA inside the particle with two salt bridges Asp312-Arg314. The two FrhA molecules are viewed from inside the complex. (**D**) The C-terminal three-stranded β-sheet in FrhA with the [NiFe] center (green and brown) and another ion (gray) near the C-terminal His386. The conserved residues Glu44, Glu229, and His386 coordinate the ion.

both of the conserved lysines may interact with the phosphate group of $F_{420}$. The long α-helix on the surface that was interpreted as the C-terminal helix of FrhB is not very well resolved. The residues in the helix are mostly not conserved in FrhB, except at the N-terminal end (*Figure 7*). In our model, the conserved residues are located near the loop 164GKGK mentioned above. This region also contains several other conserved residues (*Video 5*) and likely constitutes the access channel

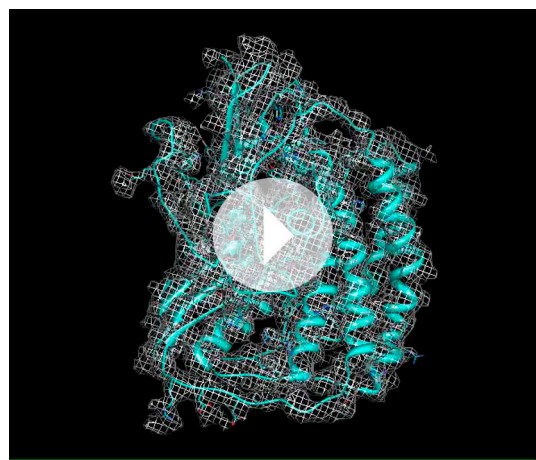

**Video 2**. FrhA. The model of FrhA is shown superimposed on the map.

for the substrate $F_{420}$. The map of Frh in the presence of substrate is similar to the substrate-free map, indicating that $F_{420}$ binding does not involve major conformational changes.

The residues coordinating the Fe-S cluster, FAD, and $F_{420}$ are conserved in the FrhB family, which includes subunits of $F_{420}$-dependent sulfite reductase (Fsr), $F_{420}H_2$:quinone oxidoreductase (FqoF), $F_{420}H_2$:phenazine oxidoreductase (FpoF), $F_{420}$-dependent glutamate synthase and formate dehydrogenase (FdhB) (*Johnson and Mukhopadhyay, 2005*) (*Figure 9*). Some of these proteins contain additional domains, like an N-terminal ferredoxin domain (which in Frh is part of FrhG), or a C-terminal domain containing a binding site for the substrate of the electron transfer from $F_{420}$. Clearly, these enzymes share a common fold with FrhB.

## Discussion

Recent advances in instrumentation and image processing procedures have made cryo-electron microscopy of isolated macromolecular complexes a powerful technique in structural biology. After the first virus structure at subnanometer resolution (*Böttcher et al., 1997*) showed the feasibility of the method, icosahedral viruses were reconstructed in recent years to better than 4 Å resolution (*Yu et al., 2008*, *2011*, *Zhang et al., 2008*, *2010a*; *Liu et al., 2010*; *Wolf et al., 2010*; *Chen et al., 2011*; *Settembre et al., 2011*; ), taking advantage of the 60-fold symmetry and huge size (~20 to 150 MDa), which together contribute to the relatively high signal-to-noise ratio for these particles that makes the determination of the orientation unambiguous. For smaller complexes, the techniques have been pioneered with the 800-kDa GroEL complex with D7 symmetry, reaching subnanometer resolution for the first time in 2004 (*Ludtke et al., 2004*) and ~4 Å in 2008 (*Ludtke et al., 2008*). Subsequent reconstructions to this resolution were achieved for archaeal (*Zhang et al., 2010b*) and mammalian (*Cong et al., 2010*) chaperonins. The chaperonins have different conformational states, and usually only one of these gave a high-resolution map, whereas for the others the resolution was limited due to conformational flexibility. The appearance of our reconstructions is similar to the chaperonins; in agreement with this, Fourier shell correlation indicates a resolution of 3.9 and 4.0 Å for the two maps, respectively (*Figure 10A*). Recently it has been noted that the resolution of such reconstructions are usually overestimated due to correlation of noise at high resolution (*Scheres and Chen, 2012*). Using a 'gold standard' procedure to estimate resolution, where two half-data sets are independently refined to a reference not containing high-resolution information (see 'Materials and methods'), yielded 5.5 Å (*Figure 10B*). However, an objective indication of (local) resolution is provided by the appearance of secondary structure: the α-helical pitch of 5.4 Å is very well defined in many regions (*Figure 2F,G*), as is the separation of β-strands (axial distance ~4.8 Å) (*Figure 5B*). Comparison of these areas with maps calculated from the model low-pass filtered to different resolutions suggest a resolution ~4.5 Å. The overall resolution estimate is an average of these well-resolved protein regions and other regions that are less well resolved, especially those at the periphery of the complex. An FSC curve between the experimental map and a map calculated from the model indicated a resolution of 5.8 Å (*Figure 10C*); this figure also constitutes an average over the whole structure, with the model probably best for internal α-helices and worst for loops at the periphery. Since the model was derived from the map, and not from an independent experiment, this measure does not properly reflect the resolution of the map, but rather describes how well our admittedly incomplete model explains the map densities. It should be noted that we used tight protein structure restraints during modeling, in order to prevent overfitting to noise while creating an unrealistic model that would have yielded a higher model-to-map correlation.

Several factors contributed to the successful structure determination of Frh. The tetrahedral symmetry means that there are 12 asymmetric units in each particle, less than the 14 and 16 in the D7 and D8 symmetric chaperonins, but the resulting ball-like shape ensures that the complex has no preferential

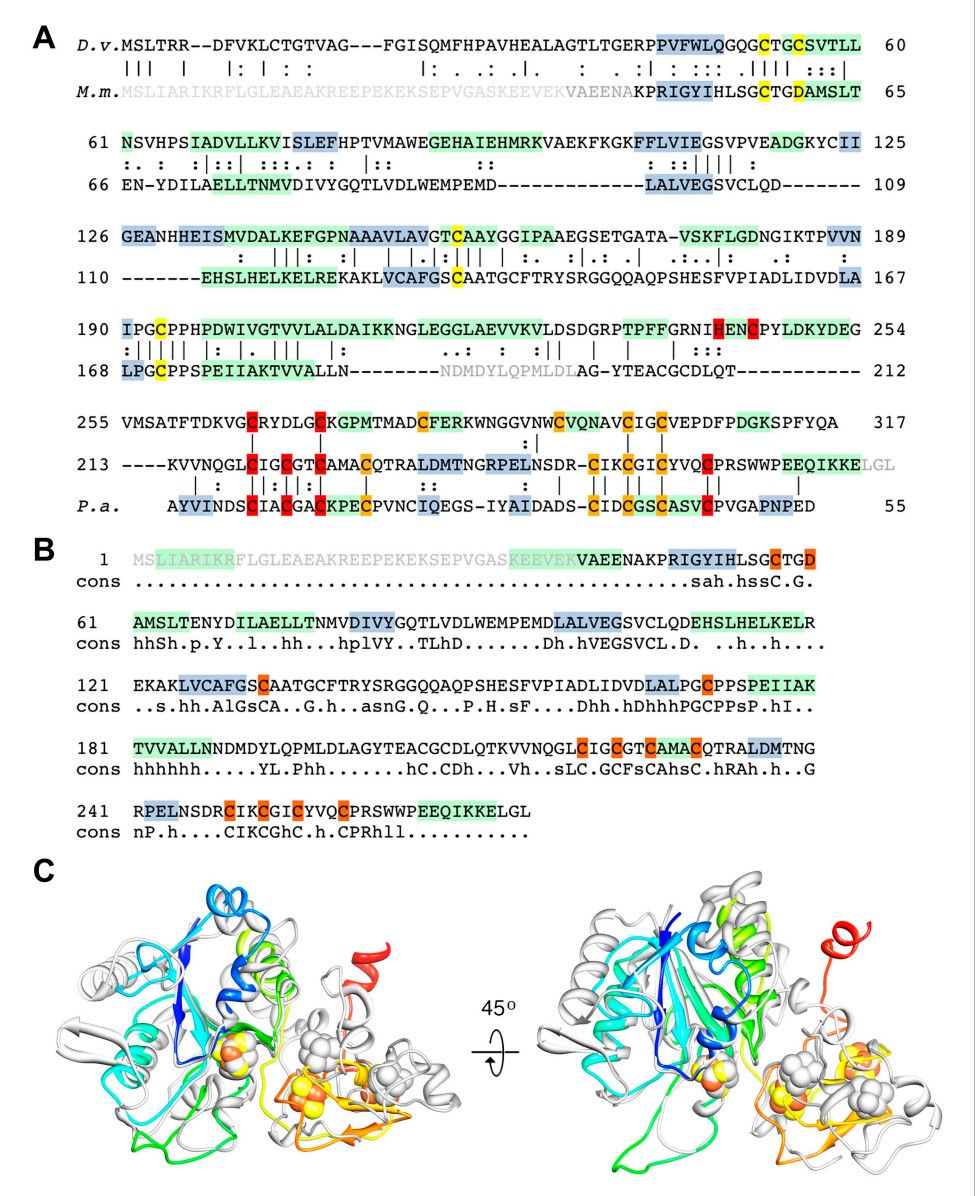

**Figure 6**. Sequence and secondary structure of FrhG and a comparison with homologous proteins. (**A**) Alignment of the [NiFe]-hydrogenase small subunit from *Desulfovibrio vulgaris* Hildenborough (first line) with *Methanothermobacter marburgensis* FrhG (third line) and ferredoxin from *Peptostreptococcus asaccharolyticus* (pdb 1dur) (last line). The second and fourth lines show the identical (|) and similar (:) amino acids. The alignment was done with ClustalW. The α-helices are highlighted in green, β-strands in blue. Ligands of the proximal [4Fe4S] cluster are in yellow, of the medial cluster in orange, and of the distal cluster in red. Amino acids not seen in the Frh structure are in gray font. The first 38 amino acids of the FrhG sequence deduced from the *M. marburgensis* genome are probably not part of the protein (see main text) and are shown in light gray font. (**B**) Consensus sequence of archaeal FrhG species. Capitals: conserved residues; lower case: similar residues (h: hydrophobic; s: small (GAS); l: large (LIFYHW); a: aromatic (FYWH); z: T or S; n: D or E; p: R or K). (**C**) Comparison of the FrhG model (rainbow coloring from blue to red) and the [NiFe]-hydrogenase small subunit from *Desulfovibrio vulgaris* Hildenborough (pdb 2wpn) (gray) in two different orientations (left and right). The structure near the proximal [4Fe4S] cluster (blue-green) is conserved, but the periphery diverges. The ferredoxin domain (yellow to red) containing the medial and the distal iron-sulfur clusters is not homologous and the clusters do not overlap.

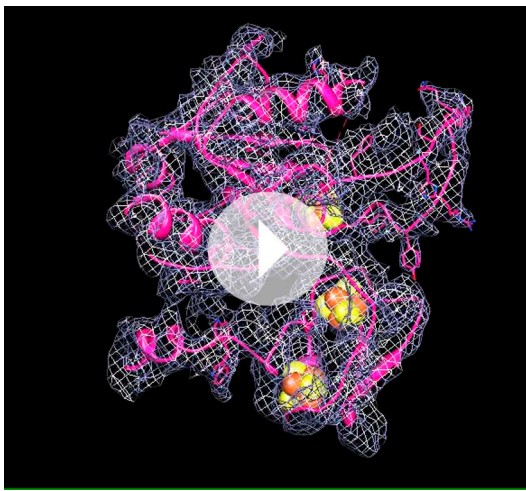

**Video 3**. FrhG. The model of FrhG is shown superimposed on the map.

orientation in the ice (*Figure 1D*), unlike the cylinder-shaped chaperonins that tend to be oriented with their symmetry axis parallel or perpendicular to the ice (*Ludtke et al., 2001*; *Zhang et al., 2011*). From the quality of the reconstructions, we conclude that the complex is very rigid. The maps of Frh with and without substrate are of similar resolution and quality, implying that any protein conformational changes due to $F_{420}$ binding are too small to detect at the present resolution. The density for $F_{420}$ is weak, but its position was inferred at an early stage of the analysis from the fact that maps of Frh without the substrate were consistently empty and maps with $F_{420}$ always showed some density here. The later tracing of the amino acid chain of FrhB and the FAD confirmed the assignment: $F_{420}$ is in van der Waals distance from its hydride donor, the FAD isoalloxazine ring, and the residues in the vicinity are highly conserved in FrhB and in the FrhB family of $F_{420}$-binding proteins (*Figures 7, 9* and *Video 5*).

While the chain trace for FrhA and FrhG was relatively straightforward using the group I hydrogenase homolog, FrhB was built ab initio. The visibility of the FeS cluster density (*Figure 2D* and *Video 1*) made it possible to use the four Fe-S ligand cysteines together with a secondary structure prediction as an initial guide; large side chain densities helped to establish the correct sequence assignment. The secondary structure prediction was fully confirmed, and weak or discontinuous densities in the map were almost invariably found to represent glycine residues, further indicating a correct assignment. Density for the FAD cofactor only became clear when most of the protein chain was placed; density for the ribose and ribitol are missing (*Figure 8*). The pyrophosphate group is closely associated with the N-terminal α-helix 26–64 of FrhB. The N-terminus of this helix is one of the most conserved regions in the FrhB family (*Figures 7, 9*), and its role in compensating the negative charge of the phosphates is a common motif in FAD-binding proteins (*Dym and Eisenberg, 2001*). Conserved regions of FrhB were found near the FAD and surrounding the $F_{420}$-binding site and its putative entrance channel (*Video 5*). Of the three proteins, only FrhG

```
  1    MVLGTYKEIVSARSTDREIQKLAQDGGIVTGLLAYALDEGIIEGAVVAGPGEEFWKPQPM
       ..hG.Y..hhsARs....h...sQDGGhhz...hYsln..hhnsshhh......a.....

 61    VAMSSDELKAAAGTKYTFSPNVMMLKKAVRQYGIEKLGTVAIPCQTMGIRKMQTYPFGVR
       hhhhhhEh..s.GzKYz.sPNh..hK.AhR.aGhn.hshhs..CQ..shRK...YP...R

121    FLADKIKLLVGIYCMENFPYTSLQTFICEKLGVSMELVEKMDIGKGKFWVYTQDDVLTLP
       .h..KI.hhhGIaCMENF.....................K..IGKGKF..........h.

181    LKETHGYEQAGCKICKDYVAELADVSTGSVGSPDGWSTVITRTDAGDSIFKQAVEAGLFE
       L..TH.YEQ.sC..C.DYhs.LsDhzzGSVGsPDGWSThh.Rz...G..ll........lE

241    TKPIEEVKPGLGLLEKLAAQKKEKAEKNIAARKEMGLPTPF
       ......hKPGL.Lh.KLA..K.....K.h.....hG.....
```

**Figure 7**. Sequence and secondary structure of FrhB. Light gray font: No density; green highlight: α-helix; gray highlight: β-strand. Second line: Consensus sequence of FrhB species. Capitals: conserved residues; lower case: similar residues (h: hydrophobic; s: small [GAS]; l: large [LIFYHW]; a: aromatic [FYWH]; z: T or S; n: negative [D or E]; p: positive [R or K]). In the consensus sequence, the residues for coordination of the iron–sulfur cluster, FAD, and $F_{420}$ are highlighted in orange, cyan, and yellow, respectively.

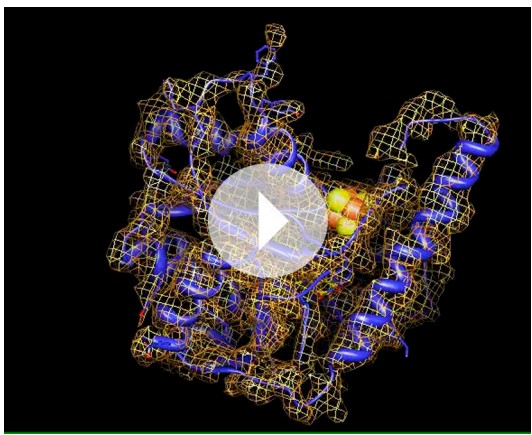

**Video 4**. FrhB. The model of FrhB is shown superimposed on the map.

is missing density for an internal stretch, residues 188–199 (*Figure 6A*). The flanking residues of this stretch are located less than 10 Å apart on the surface of the complex and they probably form a flexible loop. FrhA and FrhB were traced over their full length. All three proteins follow the rule that hydrophobic residues form the core of the protein and charged residues face the outside. We found several pockets lined with hydrophobic side chains, not just in FrhA and FrhG, where these features are shared with the bacterial protein used as a template for modeling, but also in FrhB. FrhB also contains a highly hydrophobic helix (27GIVTGLLAYAL), which is buried completely inside the subunit. These features all indicate a correct chain trace.

Our structure shows that each heterotrimer encompasses a complete electron transfer chain that runs from the [NiFe] cluster in FrhA via the three Fe-S clusters in FrhG and one Fe-S cluster and FAD in FrhB to $F_{420}$ (*Figures 4D, 11* and *Video 6*). The cofactors describe a curvilinear path in the protein interior, almost parallel to the surface of the complex, with distances of ~10 Å between neighbors, except FAD and $F_{420}$, which are in van der Waals contact to enable hydride transfer (*Figure 8C*). The predicted $F_{420}$-binding site is located on the outside of the complex, which facilitates access of the cytosolic $F_{420}$. The two electron transfer chains in a dimer are separated by at least 17 Å and are thus independent of each other, indicating that the functional unit is one

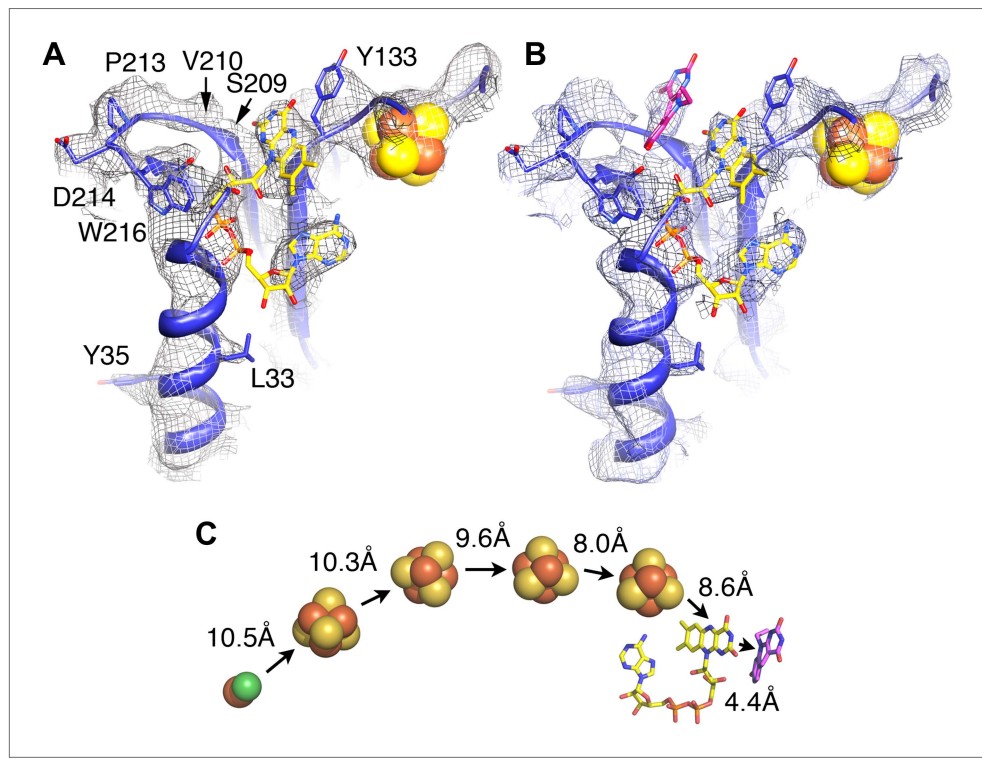

**Figure 8**. Electron-transfer chain and $F_{420}$. Maps in the absence (**A**) and presence (**B**) of the substrate $F_{420}$ differ in a region near a conserved loop between two β-strands near the FAD (carbons in yellow). The isoalloxazine ring of $F_{420}$ fitted into this density is shown in pink. (**C**) Electron transfer chain with minimal distances indicated.

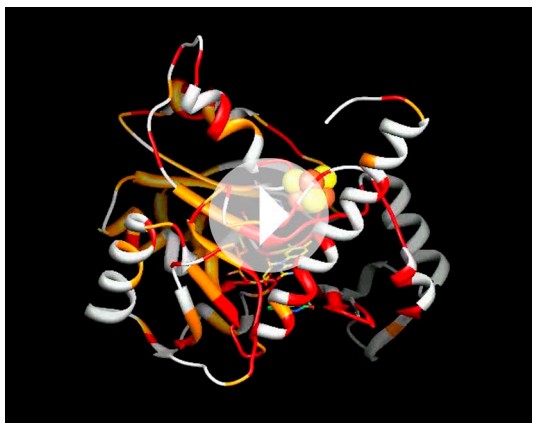

**Video 5**. Conserved residues in FrhB. The FrhB model is shown with conserved residues in red, similar residues in orange and unconserved residues in gray (see **Figure 7** for sequence). FAD is yellow and $F_{420}$ green. Highly conserved regions surround the FeS cluster and the FAD as well as the $F_{420}$ isoalloxazine ring. An entrance pathway for $F_{420}$ is suggested as well.

FrhABG heterotrimer and there is no cooperativity in the complex. The three proteins in the Frh heterotrimer form a tight complex with extensive subunit interfaces and a large number of subunit interactions, including two salt bridges between loops of the two FrhA molecules in a dimer, from Asp312 of one FrhA and Arg314 of the other (**Figure 5C**). The heterotrimer is quite slender compared to the two-subunit [NiFe] hydrogenases of known structure, with a small contact surface especially between FrhB and the other two subunits (**Figure 12**), and complex formation may be necessary for stability of the complex and keeping the metal clusters at optimal distances. The same quaternary structure of Frh has been observed not only in the closely related thermophile *M. thermautotrophicus* (**Wackett et al., 1987**) but also in the phylogenetically distant mesophilic *M. voltae* (**Muth et al., 1987**) and *M. hungatei* (**Sprott et al., 1987**), so it is clear that the formation of a large complex plays an essential role for the function of Frh.

## Materials and Methods

### Cultivation

*Methanothermobacter marburgensis* (DSM 2133) was obtained from the Deutsche Sammlung von Mikroorganismen (DSMZ, Braunschweig, Germany). The archaeon was grown anaerobically at 65°C on 80% $H_2$/20% $CO_2$/0.1% $H_2S$ in a 12-L fermenter containing 10 L complete mineral salt medium (**Schönheit et al., 1980**). Cells were harvested by the use of a continuous-flow centrifuge under anoxic conditions at the late exponential phase and stored at −80°C.

### Purification of the Frh complex from *M. marburgensis*

Purification was performed under strictly anaerobic conditions at 18°C in an anaerobic camper (Coy Laboratory Products, Grass Lake, MI). All buffers used contained 2 mM DTT and 25 µM FAD. Cell extracts were routinely prepared from 20 g (wet mass) of *M. marburgensis* cells. The cells were suspended in 35 ml 50 mM Tris/HCl pH 7.6 (buffer A), and passed four times through a French pressure cell at 125 MPa. Cell debris were removed by centrifugation at 15,000×g for 30 min. The supernatant, designated cell extract and containing ~1000 mg protein, was adjusted to 800 mM ammonium sulfate in buffer A and stirred for 20 min. The cell extract was applied to a Phenyl Sepharose 6 Fast Flow column (6 × 10 cm) equilibrated with 800 mM $(NH_4)_2SO_4$ in buffer A. Protein was eluted by a $(NH_4)_2SO_4$ step gradient in buffer A: 800 mM $(NH_4)_2SO_4$ for 250 ml, 200 mM $(NH_4)_2SO_4$ for 250 ml, and 0 mM $(NH_4)_2SO_4$ for 50 ml (flow rate: 8 ml/min). The Frh activity was eluted in the 0 mM $(NH_4)_2SO_4$ fractions. The protein solution was concentrated with ultrafiltration by Amicon filters (100-kDa cutoff) to 4–5 ml, which were then applied to a Sephacryl S-400 HR column (2.6 × 60 cm) equilibrated with buffer B (buffer A + 150 mM NaCl). The Frh activity was eluted after washing the column with 180 ml buffer B (flow rate: 1 ml/min). The pooled fractions were concentrated with ultrafiltration by Amicon filters (100-kDa cutoff) to 20 mg/ml. CHAPS was added to the concentrate (48 mM final concentration), and the solution was incubated for 12 hr at room temperature with slow stirring. The protein solution was washed 5 times on Amicon filters (100-kDa cutoff) with buffer A with 4 mM CHAPS (buffer A2) and applied to a MonoQ column (1 × 8 cm) equilibrated with buffer A2. Protein was eluted by a NaCl linear gradient in buffer A2: 0–400 mM NaCl in 25 ml and then 400–600 mM NaCl in 20 ml (flow rate: 0.8 ml/min). In the linear gradient of 400–600 mM NaCl, the active Frh was eluted from the column at a concentration of 540 mM NaCl. The protein solution was concentrated on Amicon filters (100-kDa cutoff) to 2 ml

```
Cons          -----l-clhhAps--------sQDGGhhzsll-Yhlcc--hnshhhh-------acs--hhh--s--h   68
FrhB    1     MVLGTYKEIVSARSTDREIQKLAQDGGIVTGLLAYALDEGIIEGAVVAGPG-EEFWKPQPMVAMSSDEL   62
GS    268     -EPLGEYTEILSARA----PMFRGQDGGVVTALLTYALREGIVDGALVVDRDPAMPWKPVPVLAEDPEDV   332
FqoF  114     -NGLGEYIEVVAARSKR----FVGQDGAMVTEFTASALEMGIIERAIFVARD--SNWRTRVVTIKTPEQL   177
FpoF   74     -NELANVRKFFAARS---KENAGSQDGGVTSGILKSLFKQGKIDCAVGITR--DEKWESKVVLLTSAEDV   137
Fsr    66     -IREKFYEKYYYAKS-----DIEGQDGGVVTAFLKYLLENGKIDGAIVVG---DECWKPVSLVVQNAEDL   126
FdhB    1        MKYLLARATDEEIQRKGECGGAVTAIFKYMLDKEVVDAVLTLERG-YDVYDGIPVLLEDSSGI   62

Cons          --h--zKY----—-h--L----------phshhh--PCQ--shp-hc--------          ------
FrhB   69     KAAAGTKYTFSPNVMMLKKAVRQYGIEKLGTVAIPCQTMGIRKMQTYPFGVRF-----------LADKIK   127
GS    333     VRAAGTKYSVCPILKVLKE351  1MSRYAMVGTPCQITAATLMKEYN---------------GEFPVE    29
FqoF  178     YDRKITGTKYSYADVLPALKEAVLKSEAVGFVGTPCMVSAVRKMQQAFKKFER---------------VK   232
FpoF  138     EKVRGTKYTSDPVVAALREAFEKY--DRIAVVGVPCQAHSARLIREN----------------VSEKIV   188
Fsr   127     LKTAKSKYAIST-LDALRKAGEMG-LEKVAVVGLPCQINGLRKLQYFPYHAKHDLELGRNGKPVKLPKIE   194
FdhB   63     ESTCGSLHCAPTMFGDLISRYLSD--MRLAVAVKPCDAMAIRELEKRHQ--------------IDPDKV   115

Cons          h-hGllC---l--------l------h-hc-V---cl-pGpllh-h-n-----h-lc----    -----C
FrhB  128     LLVGIYCMENFPY-TSLQTFICEKLGVSMELVEKMDIGKGKFWVYTQD-DVLTLPLKETHG---YEQAGC   192
GS     30     LRIGLFCMENFSY-TYLREL-AEAEGVDLRDVSECRIEKGRLWFHLNDGSTVSIPLERARS---AMRKNC    94
FqoF  233     LAIGLFCTENFYH-HDLYKFLLEKANADLRNAVKTDIKKGKFIVEMKDGSKVRIPVKDFEE---IIPSGC   295
FpoF  189     LIIGLLCMESFHHDVMLDKIIPEIMKVKIEDVRKMEFTKGKFWVYTSDGEVHSVPIKDVAK---YARNPC   255
Fsr   195     YLIGLFCTEKFRYDNMKEVLS--KHGIDIEKVEKFDIKKGKLLVYVN-GEKKEFDLKEFEI-----CSGC   256
FdhB  116     YKIGLNCGGTLAPVSAREMIETF-YEIDPDDVVSEEIDRGKFIVELRDGSHREISIDYLEEEGFGRRENC   184

Cons          --C--l-s--sDhshG-hGs-DGaz-h--pTn-Gp-l-c-h--------------------hcKh-----
FrhB  193     KICKDYVAELADVSTGSVGSPDGWSTVITRTDAGDSIFKQAVEAGLFETKPIEEVKPGLGLLEKLAAQKK   262
GS     95     SVCMDFTSEQSDVSVGSVGSPQGWSTLIIRTERGRELVDGAAKAGYIETAPITGK--GLKLLEKLASGKK   162
FqoF  296     KVCQDFAAVESDVSVGSVGSPNRFSTVMVRTEVAKQILDYIREKDYAEFG-----EAKLDIVQKLCDHKM   363
FpoF  256     HHCCDYTSVFADISVGSVGAPDGWNSVFIRTDAGEEYFEMVREE----MEIMEDPKPGLELVKKLIDMKR   321
Fsr   257     KMCRDFDAEMADVSVGCVGSPDGYSTIIIRTEKGEEIKNAVELK----------EGVNLEEIEKLRQLKL   316
FdhB  185     QRCEIMVPRNADLACGNWGADDGWTFIEVNTERGQEIIEGARSSGYIE------AREPSEKMVKIREKIE   248

Cons          -------------------
FrhB  263     EKAEKNIAARKEMGLPTPF                                                     281
GS    163     EENLEEIQRRESVARPVLYWRVMPGDLYPEEIKDCQFDDLRADVIDVGACVLCGACEASCPEGIVRIND-   231
FqoF  364     KIHPWPPKKKEEKSEE                                                        379
FpoF  322     KNNAEHFKEVCKEFSFETGIRDETV                                               346
Fsr   317     KRFKKEVERRRENNEYVSFYWTADYGGIGKRADGTYFIRVRAKPGGWYKPEEIKEILDIAEEYNAKIKV-  385
FdhB  249     NAMISMARKFQDKYLDEEYPSLDEWDEYWKRCINCFACRDACPVCFCRECELEKDYLLESDEKAPDPLT-  317
```

**Figure 9**. Alignment of the FrhB family of F$_{420}$-binding proteins. FrhB: FrhB of *Methanothermobacter marburgensis* DSM 2133; GS: F$_{420}$-dependent glutamate synthase from *M. marburgensis*, ADL58239; FqoF: F$_{420}$H$_2$:quinone oxidoreductase subunit F from *Archaeoglobus fulgidus* DSM 4304, NP_070660; FpoF: F$_{420}$H$_2$:phenazine oxidoreductase subunit F from *Methanosarcina barkeri* str. Fusaro, YP_303819; Fsr: N-terminal domain of Fsr, F$_{420}$-reducing sulfite reductase from *Methanocaldococcus jannaschii* DSM 2661, Y870_METJA; FdhB: beta subunit of formate dehydrogenase from *M. marburgensis*, YP_003850414. Cons: the consensus sequence of the FrhB family. Capitals: conserved residues; lower case: similar residues (h: hydrophobic; s: small [GAS]; l: large [LIFYHW]; a: aromatic [FYWH]; z: T or S; n: negative [D or E]; p: positive [R or K]; c: charged [D, E, R, K]). In the consensus sequence, residues indentified in the FrhB structure for coordination of the iron–sulfur cluster, FAD, and F$_{420}$ are highlighted in orange, cyan, and yellow, respectively. Secondary structure prediction for each of the proteins was done with PSIPRED (http://bioinf.cs.ucl.ac.uk/psipred/): green highlight: α-helix; gray highlight: β-strand. Highly conserved amino acids are indicated in bold font. Amino acid numbers are shown in red; note that the C-terminal 83 residues of glutamate synthase (GS) align with the N-terminus of FrhB and the N-terminal 182 residues with the C-terminus of FrhB.

and further purified on a Sephacryl S-400 HR column in order to remove remaining smaller particles and bigger aggregates.

F$_{420}$ was isolated from *M. marburgensis* by established methods (**Shima and Thauer, 2001**).

## Cryo-electron microscopy

3 µl of a 0.7 mg/ml Frh sample was applied to freshly glow discharged Quantifoil R1/4 grids (Quantifoil Micro Tools, Jena, Germany) that had been pretreated for 15 s in chloroform. The |grids were blotted in an FEI Vitrobot using a 2.5 s blotting time at 70% humidity and 10°C and plunge-frozen in liquid ethane. For collection of data of Frh in the presence of substrate, 0.5, 1.0, or 10 mM F$_{420}$ was added to the sample under oxygen-free conditions just prior to grid preparation.

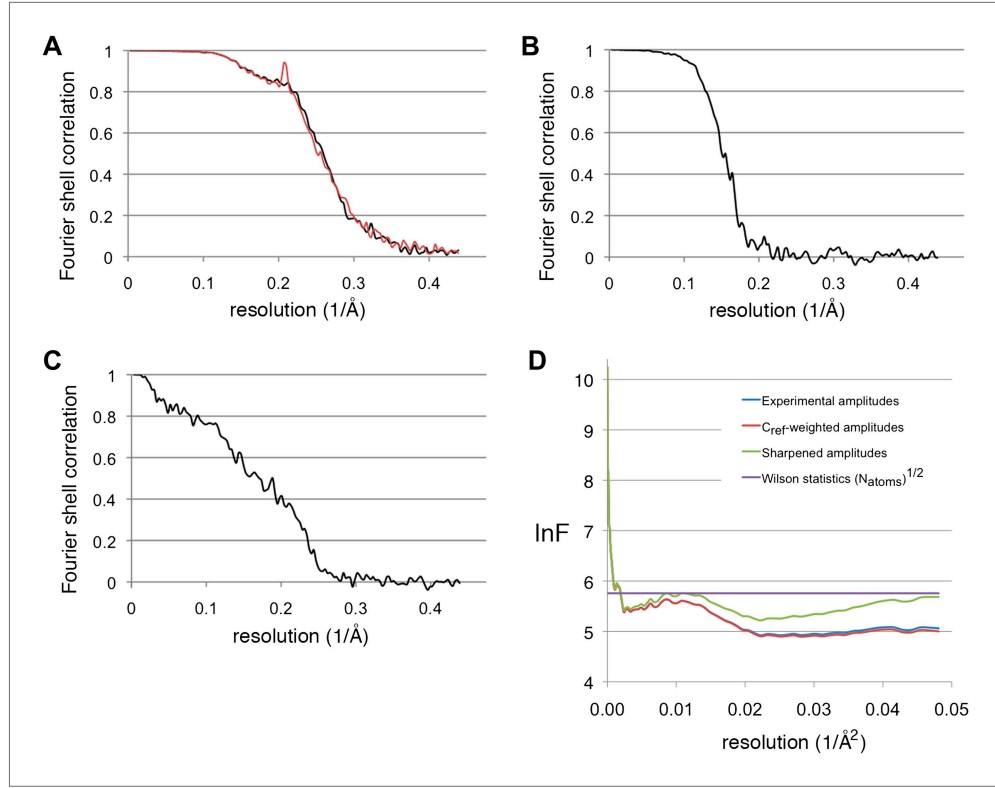

**Figure 10**. Resolution estimations and B-factor determination. (**A**) Fourier shell correlation plot, showing the resolution for the substrate-free map (black) and the $F_{420}$-containing map (red). At the 0.5 FSC criterion, the resolution is 3.9 and 4.0 Å, respectively. (**B**) Gold standard FSC between two half-data sets independently refined from a low-resolution model. The resolution at the 0.143 FSC criterion is 5.5 Å. (**C**) FSC between the map and model. At 0.5 FSC, the indicated resolution is 5.8 Å. (**D**) Plots of the natural logarithm of the spherically averaged structure factor amplitude as a function of the resolution (Å$^{-2}$). Blue: experimental amplitudes; red: $C_{ref}$-weighted amplitudes; green: amplitudes sharpened with a B-factor of −54 Å$^2$, as determined by the program *embfactor* (***Fernández et al., 2008***) for the resolution zone 10–4.5 Å. The purple line shows the average scattering amplitude for the Frh complex ($\sqrt{N_{atoms}}$).

Images were collected at liquid nitrogen temperature on an FEI Tecnai Polara operated at 200 kV. Before images were recorded, the microscope was carefully aligned in an iterative process to correct for objective astigmatism and beam tilt by coma-free alignment (***Glaeser et al., 2011***). The corrections were carried out at a dose of 15 e⁻/Å$^2$ and at half the defocus value used for collecting the images, and were repeated for each grid square from which images were collected. The alignments were done with a Gatan 4k × 4k CCD using unbinned images.

Images were recorded on Kodak SO-163 film at a magnification of 59,000× with a dose of 10–15 e⁻/Å$^2$ at a defocus of 1.5–2.8 μm. The film was developed for 12 min in full-strength Kodak D-19 developer and fixed for 8 min in Kodak Rapid Fix. Five hundred and six negatives were collected of substrate-free Frh and 512 of Frh with $F_{420}$. Films that showed obvious flaws (too thin ice with particles just around the edge of the Quantifoil hole, too high particle concentration, broken ice, obvious drift) were discarded. Four hundred and two negatives of substrate-free Frh were scanned and 277 of $F_{420}$ with Frh. The latter preparation had a higher protein concentration, resulting in many films with too many overlapping particles. This was more than compensated by the higher number of particles per film.

## Image processing

Films were digitized on a Zeiss Photoscan scanner with a pixel size of 7 μm, corresponding to 1.14 Å on the specimen as calibrated with fatty acid synthase (***Gipson et al., 2010***). Particle selection

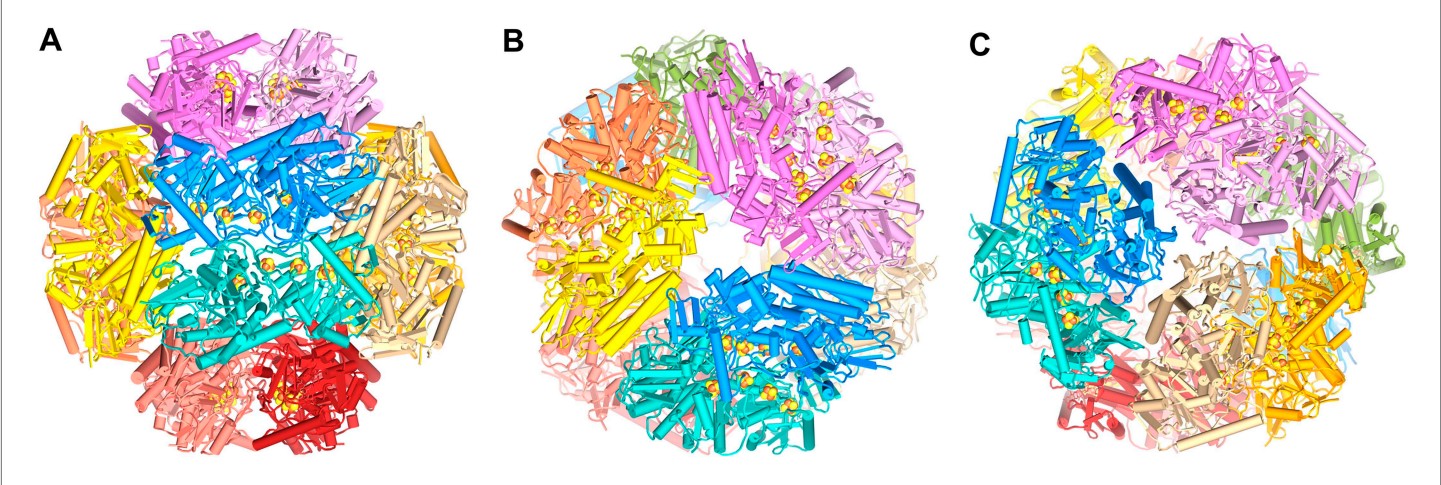

**Figure 11**. Model of the dodecameric Frh complex. Each FrhABG heterotrimer is colored differently with the same color scheme as the map in **Figure 1**. (**A**) View down the twofold axis, (**B**) view of the FrhB trimer, and (**C**) view of the closely packed FrhA trimer as **Figure 1C**.

was done semiautomatically with the Boxer module from EMAN (**Ludtke et al., 1999**) and data processing with EMAN2 (**Tang et al., 2007**). The contrast transfer function (CTF) of the selected particles from each film was determined with EMAN2 and images that showed visible Thon rings in the power spectrum to high resolution and no indication of drift or astigmatism were selected for further processing and their CTF was corrected by phase flipping. A tetrahedral starting model was created by the EMAN2 program *e2initialmodel* from a number of class averages. This model was iteratively refined using the main EMAN2 program *e2refine,* which determines the 3D orientation of each particle by comparison to a set of projections of the current 3D reference map. Particles in the same orientation are aligned and averaged, and a new 3D map is constructed from the averages that are then reprojected to create references for the next refinement cycle (**Ludtke et al., 1999**; **Tang et al., 2007**). Tetrahedral ($T$) symmetry was applied throughout. In the initial refinement steps, the data were binned to a pixel size of 2.28 Å. After the resolution reached ~10 Å, the unbinned data were used. At every refinement step, the 30% worst members of each class were discarded. Because of the high symmetry and globular shape of the particle, all class averages had similar quality (see **Figure 1B**) and were used for each reconstruction. The final data set of substrate-free Frh contained 84,000 particles from 101 negatives of the 402 scanned negatives. Between iterations, the projection angle was varied. In the last refinement step an angle of 0.9° between consecutive reference reprojections was used, yielding 2134 reference images.

The $F_{420}$ data were refined with a high-resolution map of substrate-free Frh as a starting reference map. Ninety seven thousand particles were selected from 80 negatives. Smaller datasets of Frh with 0.5, 1.0, and 10 mM $F_{420}$ were used initially, but no significant differences were seen and the datasets were merged.

The resolution of the maps was estimated using the command *eotest* in EMAN2, which calculates the Fourier shell correlation (FSC) between

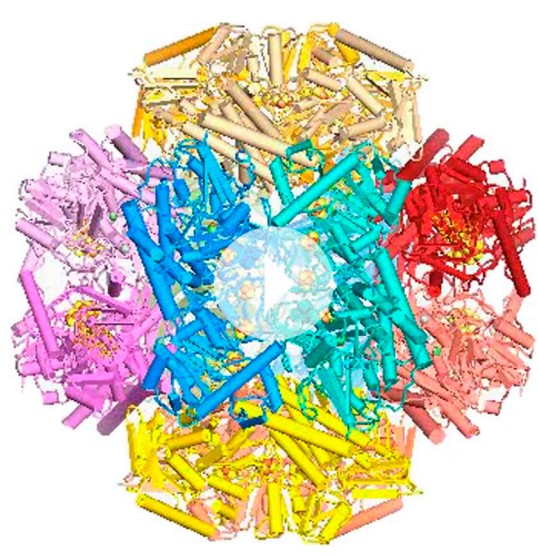

**Video 6**. The Frh dodecamer. A model of the tetrameric Frh complex.

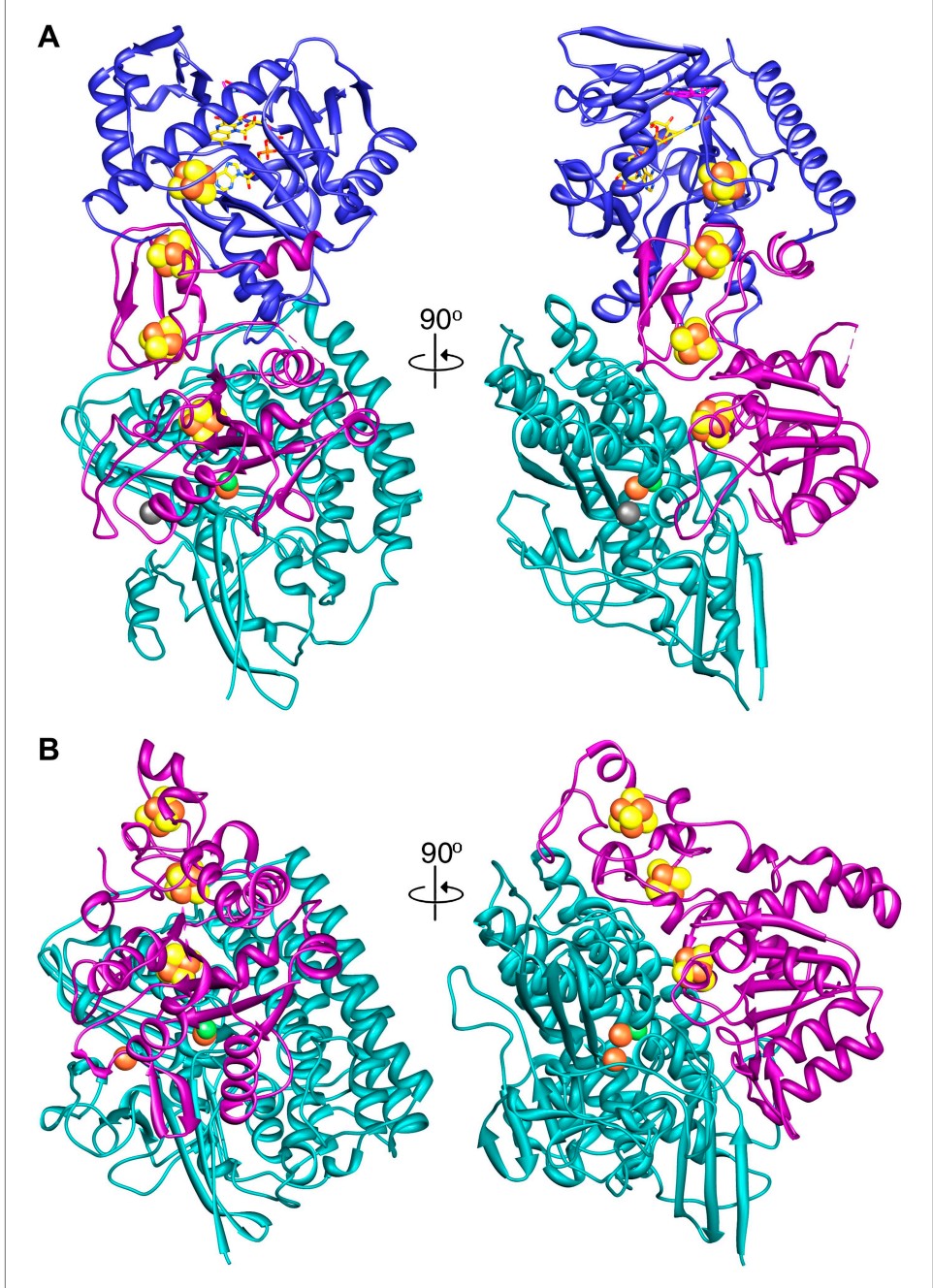

**Figure 12**. A comparison of (A) Frh and (B) the group I [NiFe] hydrogenase from *Desulfovibrio gigas* (2wpn) (***Marques et al., 2010***) shows a lower intersubunit contact area in the former. The large subunit/FrhA is shown in green, the small subunit/FrhG in magenta, and FrhB in blue. The left panel shows Frh in an orientation along the twofold axis of the complex, as seen from the outside, and the right panel a view 90° rotated, as seen from the dimer partner.

reconstructions made by half-data sets from the odd and even numbered particles. This indicated 3.9 and 4.0 Å for the maps without and with $F_{420}$, respectively, using the 0.5 FSC criterion (***Figure 10A***). The resolution estimate can be inflated by the overfitting of noise at high resolution (***Grigorieff, 2000***; ***Scheres and Chen, 2012***). To test for this effect, we used the procedure *e2refine-evenodd* in EMAN2, where the phase of the starting model are randomized from a cutoff resolution lower than the expected

resolution of the map, and then two half-data sets are refined to convergence against this model. High-resolution noise will be uncorrelated between the two maps. Subsequently, the FSC between the two resulting maps is calculated, and the resolution from these two independent half-data sets determined at 0.143 FSC (*Rosenthal and Henderson, 2003*). This procedure yielded a resolution of 5.5 Å (*Figure 10B*). B-factors of the reconstructions were estimated with the program *embfactor* (*Rosenthal and Henderson, 2003*; *Fernández et al., 2008*) in the resolution range 10–4.5 Å as 54 $Å^2$ for the substrate-free map and 60 $Å^2$ for the map with $F_{420}$ (see *Figure 10D*).

## Model building

Maps were visualized in Chimera (*Pettersen et al., 2004*). Homologous protein structures were fitted manually in the map followed by a rigid body fit (pdb 2wpn, *Marques et al., 2010*, for FrhA and the C-terminal domain of FrhG; pdb 1dur for the ferredoxin domain of FrhG). Models were then mutated to the correct amino acid and manually rebuilt in Coot (*Emsley and Cowtan, 2004*). Regions for which no homolog was available, including all of FrhB, was built ab initio in Coot, based on secondary structure predictions done using the PSIPRED server (*Bryson et al., 2005*). The conserved cysteines serving as ligands for the Fe-S clusters and the visibility of the Fe-S clusters in the map gave clear initial hints to the identity of secondary structure elements. Helices were built using the command *Place helix*. Loops and β-sheets were built by first placing $C_\alpha$ atoms using the *Baton build* module. Full-atom models were built based on the visibility of side chains, which led in all cases to an unambiguous assignment. The models were refined using the *Regularize Zone* module. Where possible, side chains were fitted to available density (overall about 55% of side chains were visible, including most aromatic residues and arginines). During refinement, torsion angle, planar peptide, and Ramachandran restraints were used in order not to create a well-fitting but unrealistic model. The final model contains 878 residues out of a possible 903, a [NiFe] cluster, 4 [4Fe4S] clusters, an FAD and part of an $F_{420}$ molecule; 88.1% of residues lie in the most favored regions of a Ramachandran plot, 8.0% in generously allowed regions, and 3.8% are outliers. A full dodecamer model was generated in Chimera (*Pettersen et al., 2004*) and a map was calculated from this model. An FSC curve between this map and the experimental map indicated an overall agreement of 5.8 Å at 0.5 FSC (*Figure 10C*).

Figures were made using Chimera (*Pettersen et al., 2004*) and the PyMOL Molecular Graphics System (*DeLano, 2002*).

## Acknowledgements

We thank Werner Kühlbrandt and Rolf K Thauer for support and comments on the manuscript and Özkan Yildiz for help with the computing facilities. We are also indebted to Ulrich Ermler for critical reading of the manuscript.

## Additional information

### Funding

| Funder | Author |
| --- | --- |
| Max Planck Society | Deryck J Mills, Stella Vitt, Mike Strauss, Seigo Shima, Janet Vonck |
| PRESTO program, Japan Science and Technology Agency (JST) | Seigo Shima |

The funders had no role in study design, data collection and interpretation, or the decision to submit the work for publication.

### Author contributions

DJM, Prepared the cryo-EM samples, collected the EM data and processed data; SV, Purified the protein, interpreted the structure and wrote the paper with contributions from all authors; MS, Processed data; SS, Initiated the project, interpreted the structure and wrote the paper with contributions from all authors; JV, Processed the data, built the model, interpreted the structure and wrote the paper with contributions from all authors

# Additional files

## Major datasets

The following datasets were generated:

| Author(s) | Year | Dataset title | Dataset ID and/or URL | Database, license, and accessibility information |
|---|---|---|---|---|
| Mills DJ, Vitt S, Strauss M, Shima S, Vonck J | 2013 | F420-reducing [NiFe] hydrogenase Frh | EMD-2096; http://www.ebi.ac.uk/pdbe-srv/emsearch/atlas/2096_summary.html | Publicly available at the Electron Microscopy Data Bank (http://www.ebi.ac.uk/pdbe/emdb/). |
| Mills DJ, Vitt S, Strauss M, Shima S, Vonck J | 2013 | F420-reducing [NiFe] hydrogenase Frh with bound substrate | EMD-2097; http://www.ebi.ac.uk/pdbe-srv/emsearch/atlas/2097_summary.html | Publicly available at the Electron Microscopy Data Bank (http://www.ebi.ac.uk/pdbe/emdb/). |
| Mills DJ, Vitt S, Strauss M, Shima S, Vonck J | 2013 | Cryo-EM structure of the F420-reducing NiFe-hydrogenase from a methanogenic archaeon with bound substrate | 3zfs; http://www.rcsb.org/pdb/search/structidSearch.do?structureId=3zfs | Publicly available at the RCSB Protein Data Bank (http://www.rcsb.org/pdb/). |

The following previously published datasets were used:

| Author(s) | Year | Dataset title | Dataset ID and/or URL | Database, license, and accessibility information |
|---|---|---|---|---|
| Liesegang H, Kaster A-K, Wiezer A, Goenrich M, Wollherr A, Seedorf H, Gottschalk G, Thauer RK | 2010 | Complete Genome Sequence of Methanothermobacter marburgensis, a Methanoarchaeon Model Organism | CP001710; http://www.ncbi.nlm.nih.gov/nuccore/CP001710 | Publicly available at GenBank. |
| Adman ET, Sieker LC | 2000 | The 2[4Fe-4S] Ferredoxins | 1dur; http://www.rcsb.org/pdb/explore/explore.do?structureId=1dur | Publicly available at the RCSB Protein Data Bank (http://www.rcsb.org/pdb/). |
| Marques MC, Coelho R, De Lacey AL, Pereira IAC, Matias PM | 2010 | The three-dimensional structure of [NiFeSe] hydrogenase from Desulfovibrio vulgaris Hildenborough: a hydrogenase without a bridging ligand in the active site in its oxidised, "as-isolated" state. | 2wpn; http://www.rcsb.org/pdb/explore/explore.do?structureId=2wpn | Publicly available at the RCSB Protein Data Bank (http://www.rcsb.org/pdb/). |

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
