## [Decision Letter]

Thank you for choosing to send your work entitled “De novo modelling of the F420-reducing [NiFe]-hydrogenase from a methanogenic archaeon by cryo-electron microscopy” for consideration at *eLife*. Your article has been favorably evaluated by a Senior editor and 3 reviewers, one of whom is a member of our Board of Reviewing Editors. The following individuals responsible for the peer review of your submission want to reveal their identity: Wes Sundquist (Reviewing editor) and Richard Henderson (peer reviewer).

The Reviewing editor and the other reviewers discussed their comments before we reached this decision, and the Reviewing editor has assembled the following comments based on the reviewers' reports.

The authors describe a cryoEM reconstruction of the F420 reducing NiFe hydrogenase from the methanogenic archaeon, *Methanothermobacter marburgensis* (both with and without bound F420 substrate). The enzyme is shown to form a spherical protein shell (Td symmetry) that contains 12 copies of the heterotrimeric building block (correcting previous models for the assembly). Two of the trimer subunits have counterparts on the class I hydrogenases, but one subunit, which is unique to the group 3 enzymes, is a novel iron-sulfur flavoprotein. The new mechanistic insights derived from the structure are limited, but the system is important and the technical achievements are impressive. The results are appropriate for publication in *eLife* if the following important issues can be addressed adequately.

Our main concerns surround the quality of the density maps and the true resolution of the reconstruction. While it is not useful to argue about a single number, a resolution that is significantly lower than 3.9 Angstroms will draw into question the validity of the model building, especially the de novo model of FrhB.

1. Since the authors have a density map (or maps) and an atomic model (with C-alphas deposited in PDB) that purports to explain the density, we ask them to present an objective, quantitative comparison (cross-correlation) between the map and the model, to back their presentation of a few local regions of the structure (e.g. Figures 2G, 5C, 5D, 8A, 8B). This should be a Fourier Shell Correlation as a function of resolution, and the authors should be careful not to introduce too many degrees of freedom in the fitting of the model into the map. If their resolution claim of 3.89 Å is correct, then this FSC between map and model should extend to a higher resolution than the FSC between the two halves of the experimental data shown in Figure 1D. It is now well known that many single particle procedures are susceptible to overfitting of noise, which gives rise to inflated resolution claims. The map/model FSC gives an objective and quantitative evaluation of the success of the entire procedure and does not easily provide misleading indications.

Alternative ways to assess resolution without bias were discussed at the last 3DEM Gordon Conference, and a paper published by Scheres & Chen this year (Nature Methods 9, 853–854) is of particular relevance. We ask the authors to follow one of the recipes described in this paper to provide a “gold-standard” Fourier shell correlation (FSC) curve.

To be clear, we would like to see both a map/model FSC and a gold standard FSC, and not just one of them.

2. It is our expectation that the revised manuscript will state a lower resolution. It will then be important that the authors make a strong case of why the map is still of sufficient quality to build a reliable de novo model. The authors can consult the EMDB database for published reconstructions at similar resolution, such as the rotavirus VP6 protein (EMDB-1461), which is a particularly well-established case since there is a direct comparison with a map at the same resolution obtained by X-ray crystallography.

---

## [Author Response]

1. The main concern of the reviewers is the quality of our map. We recognize that the figures in the original manuscript did not do justice to the map. Following a comment by the reviewers, we have estimated the B-factors of the maps and used these to sharpen the maps. This improved the map appearance greatly. We have included new figures of the sharpened maps in Figure 2, of larger regions than before, and showing many more side chains. We anticipate the reviewers will now be convinced of the validity of our model.

The appearance of the map is not unlike the icosahedral map in X. Zhang et al., 2008, which the reviewers ask us to compare, although clearly not as good as their 13-fold averaged map that was estimated as 3.8 Å resolution by comparison with an x-ray map. Comparison with a map calculated from our model at different resolutions suggest a resolution of ∼4.5 Å for the best regions.

We have carried out the suggested calculations and added the results to the manuscript, including a new figure (Figure 10) showing the different Fourier shell correlation plots.

For the gold standard FSC we have used the procedure now available in EMAN2, the program used for refinement. This procedure, *e2refine-evenodd,* randomizes the phases of the starting model from a user-defined cut-off resolution and then refines two half-data sets against this model. Afterward the FSC between the two resulting maps is calculated. As there are features in our map indicating better than 5-Å resolution (separation of β-strands) we chose a cut-off resolution of 6 Å; the FSC between the resulting half-data maps reached 5.5 Å at 0.143 FSC.

Our model was refined using torsion angle, planar peptide, and Ramachandran restraints and thus did not use many degrees of freedom. The map-model FSC was determined as 5.8 Å at 0.5 FSC, with positive correlation to ∼4 Å. Since this concerns a model derived from our EM map and not an independently determined x-ray model, as was the case in the Zhang and Scheres & Chen papers, this number is not a reliable measure of the quality of the map or the “correctness” of the model. The majority of the helices show groves with 5.4 Å spacing, and β-strands are clearly resolved in the best parts of the map (see Figure 5) showing that the local resolution in these map regions is evidently better than 4.8 Å. The overall resolution estimate is therefore an average of these well-resolved protein regions and other regions that are less well resolved, especially those at the periphery of the complex. We expect that a resolution test in which only the best map regions are considered would indicate significantly better resolution. However, this would only serve to provide a better number, without improving the map or aiding its interpretation. The correctness of our trace can only be judged by a detailed analysis of the model, as described below.

2. The rotavirus map (X. Zhang et al., 2008) indeed provides a good comparison. The authors estimated the map of the icosahedrally (60-fold) averaged and then 13-fold averaged VP6 protein as 3.8 Å by comparison with an x-ray map. The FSC of this map indicated 4.5/4.1 Å at FSC 0.5 and 0.143, respectively. The same map with the icosahedral but not the 13-fold symmetry applied showed 6.5 resp. 5.1 Å. Our 12-fold symmetric, tetrahedral map indicates 6.7 resp. 5.5 Å with the gold-standard procedure, which is not dramatically different from the icosahedrally averaged VP6 protein. As mentioned above, we estimate the best regions of our map as ∼4.5 Å by comparison with a map calculated from the model.

There are many indications that our tracing is correct, involving either features of the model that agree with known general protein structure properties or specific properties of the Frh subunits.

A. As described in the text, conserved residues of FrhB and the FrhB family of F_420_-binding proteins are found near the cofactors.

B. Similarly, large side chains fall in densities that occur in every map and cannot be due to noise.

C. All predicted secondary structure elements were confirmed in the final model. In the case of FrhB, this was not the case for an earlier incorrect tracing based on a lower-resolution preliminary map; after noticing the mistake all helices and strands were accounted for, giving us confidence in the tracing.

D. All three proteins follow the rule that hydrophobic residues form the core of the protein and charged residues face the outside. We found several pockets lined with hydrophobic side chains, not just in FrhA and FrhG, where these features are shared with the bacterial protein used as a template for modelling, but also in FrhB. FrhB also contains a highly hydrophobic helix (27GIVTGLLAYAL), which is buried completely inside the subunit.

E. At some places there are breaks in extended protein stretches. After the whole complex was fitted, we noticed that most of these breaks coincide with a glycine residue (FrhA: G15, G17, G42, G217, G275, G322, G329; FrhG: G50, G56; FrhB: G44, G51, G71, G98).

F. One of the helices of the four-helix bundle has a pronounced kink. This helix was interpreted as FrhA 166–193 and a proline residue (Pro180) was found exactly at the position of the kink.

G. In the ferredoxin domain of FrhG residues 235–244 form a β-hairpin with Gly240 at the turn. The two β-strands were predicted and the hairpin is clearly visible in the density.

H. A predicted three-stranded β-sheet in FrhB (148–172) fits nicely in density. The only conserved region of this sheet, 163IGKGK, forms a turn close to the position of the isoalloxazine ring of F420. The location suggests that one or both of the lysine residues may interact with the F420 phosphate group.

I. The long α-helix on the surface that was interpreted as the C-terminal helix of FrhB is not very well resolved. The residues in the helix are mostly not conserved in FrhB, except at the N-terminal end. In our model, these residues are located at the access channel of the substrate F420 (near the loop 164GKGK mentioned above) and may be involved in substrate binding.

All the points mentioned above have been integrated in the manuscript, if they were not there already, either in the Results or the Discussion, where they were most appropriate. We have also added a movie (Video 5) showing the location of conserved residues in the de novo traced molecule, FrhB.